# Health literacy among patients with non-communicable diseases at a tertiary level hospital in Nepal- A cross sectional study

Hari Joshi [1], Bhoj Raj Kalauni [1], Kiran Bhusal [2], Rabindra Bhandari [1], Aastha Subedi [1], Buna Bhandari [1,3]*

1 Central Department of Public Health, Institute of Medicine, Tribhuvan University, Kathmandu, Nepal,
2 Maharajgunj Medical Campus, Institute of Medicine, Tribhuvan University, Kathmandu, Nepal,
3 Department of Global Health and Population, Harvard T H Chan School of Public Health, Boston, Massachusetts, United States of America

* buna.bhandari@gmail.com

**Data Availability Statement:** All relevant data are within the manuscript and its Supporting information files. Data set are provided in supplementary files.

## Abstract

Health literacy (HL) is crucial in achieving the Sustainable Development Goal of reducing one-third of premature mortality by 2030 from Non-Communicable Diseases (NCDs) and improving Universal Health Coverage. Low health literacy is linked to poor health outcomes, and evidence shows that levels of limited HL are high, even among highly educated individuals. This study aims to assess HL levels and related factors among patients with NCDs at Tribhuvan University Teaching Hospital (TUTH) in Nepal. A cross-sectional survey was conducted at TUTH among 303 patients with NCDs with Cardiovascular Diseases, Chronic Obstructive Pulmonary Disease, Diabetes Mellitus, Hypertension, Epilepsy, Asthma and Cancer who came for follow-up from December 2022 to February 2023. Data was collected via face-to-face interviews by the trained enumerators using a structured Health Literacy Questionnaire (HLQ) containing 44 items (divided into nine domains). Multivariate logistic regression analysis was performed using SPSS version 26, with statistical significance at 0.05, to determine the associated factors of HL. The mean ±SD age of the respondents was 47.4±16.18 years. More than half of the respondents were female (56.1%). The patients had higher HL in all HL domains except 'Navigating the healthcare system'. Educational status was significantly associated with six out of nine HL domains. Co-morbidity, attendance at health-related seminars, regular physical activity, and social connectedness were associated with at least one of the domains of HL. This study identified the important factors of HL, such as socio-demographic and medical factors among patients with NCDs. This highlights the need for a comprehensive approach to address identified gaps in HL, considering its multifaceted and composite nature and promoting interventions to improve HL in high-risk populations.

## Introduction

The world is striving towards achieving the Sustainable Development Goal (SDG) target 3.4 to 'reduce by one-third premature mortality from non-communicable diseases (NCDs) by the

**Funding:** The author(s) received no specific funding for this work.

**Competing interests:** The authors have declared that no competing interest exist.

**Abbreviations:** COVID-19, Corona Virus Disease of 2019; CVD, Cardiovascular Disease; HL, Health Literacy; HLL, Health Literacy Level; HLQ, Health Literacy Questionnaire; LMIC, Low-and Middle-Income Countries; NCDs, Non-communicable diseases; SDG, Sustainable Development Goal; TUTH, Tribhuvan University Teaching Hospital.

year 2030' [1]. NCDs account for 74% of the global mortality, with 17 million deaths occurring before the age of seventy. Of these premature deaths, 86% are concentrated in low-and middle-income countries (LMICs) [1]. Authorities in the last two decades have been unable to progress significantly and attain the global voluntary NCD targets. The complexities in addressing NCD services were further exacerbated by the COVID-19 pandemic, as evidenced by the high burden among NCD patients [2]. The rapid upsurge of misinformation and disinformation spread through social media has impeded initiatives aimed at preventing and managing NCDs, leading to the amplification of health disparities on a global scale [3].

Health literacy (HL) is defined as 'the cognitive and social skills which determine the motivation and ability of individuals to gain access to, understand and use information in ways which promote and maintain good health' [4]. The World Health Organization (WHO) has emphasised the significance of HL in achieving SDG 3 and its importance in enhancing Universal Health Coverage (UHC) [5] as HL comprises various skills that influence the capability of patients to assess health care and use the healthcare system [6]. HL is fundamental in reducing the impact and lowering the risk of NCDs. The majority of premature deaths from NCDs, accounting for more than one-fourth of all fatalities worldwide, may have been prevented by enhancing HL, which enables individuals to take an active role in their health [7, 8]. HL is an important determinant of people's health condition, affecting health status and demonstrating social hierarchy [9].

A systematic review of the literature indicates that the elderly population with low levels of HL experiences worse health status, resulting in increased mortality [10]. Additionally, an inadequate HL among hospitalised patients is independently associated with adverse outcomes, such as higher hospital readmission rates [11]. In Nepal, only 27% of chronic disease patients had adequate HL [12], while 61% of university students reported having limited HL [13]. Similarly, a study conducted in the provinces of Calabria and Sicily revealed that only 35% of the Mediterranean general population had adequate HL [14]. In Europe, a population-based study on HL revealed that 47% of the individuals had limited HL [15]. Surprisingly, general education level doesn't reliably predict an individual's HL; even those with postgraduate qualifications were found to have poor HL skills [16].

However, to our knowledge, there is limited information about the HL of individuals with NCDs in Nepal. This study assessed the Health Literacy Level (HLL) and associated factors among patients with NCDs visiting a tertiary-level health centre in Nepal.

## Materials and methods

### Study design and site

A cross-sectional study was conducted at Tribhuvan University Teaching Hospital (TUTH) from December 2022 to February 2023. TUTH is one of the pioneering and largest tertiary-level hospitals in Nepal. It provides integrated clinical services, outpatient services, education, and research facilities to around 2500 outpatients daily. People from all over Nepal visit TUTH to receive various healthcare services.

### Study population

The study population for this study were patients with NCDs- Cardiovascular Diseases (CVDs), Chronic Obstructive Pulmonary Disease (COPD), Diabetes Mellitus (DM), Hypertension, Epilepsy, Asthma and Cancer. Patients aged 18 years and above with NCDs who provided informed consent and were currently undergoing follow-up care at the outpatient department of TUTH were recruited for this study. However, patients with cognitive

impairments or hearing disabilities were excluded from the study as it limits their capability to participate in the study.

## Sample size and sampling technique

The sample size was calculated using the one-proportion formula $n = Z^2 p q / d^2$, with the following assumptions: p = 0.27 (12), q = 0.73, and z = 1.96 at 95% CI and d = 0.05. Assuming a 10% non-response rate, the sample size was 334. A purposive sampling technique was used to select study participants, as it was not feasible to establish a sampling frame for the random sampling methods in the hospital's outpatient department setting.

## Data collection tools and techniques

HL was measured using a 44-item multi-dimensional Health Literacy Questionnaire (HLQ) after receiving approval from Swinburne University of Technology [17]. This questionnaire, developed by Osborne and the team, evaluates nine aspects of HL and has also been validated in Nepal [18]. The nine domains of HLQ include:

1. Feeling understood and supported by healthcare providers (4 items)

2. Having sufficient information to manage my health (4 items)

3. Actively managing own health (5 items)

4. Social support for health (5 items)

5. Appraisal of Health Information (5 items)

6. Ability to actively engage with healthcare providers (5 items)

7. Navigating the healthcare system (6 items)

8. Ability to find good health information (5 items)

9. Understanding health information well enough to know what to do (5 items) [17].

Participants in the study were asked to indicate their level of agreement with statements in domains 1 to 5 using a scale where 1 = strongly disagree, 2 = disagree, 3 = agree, and 4 = strongly agree. For domains 6 to 9, participants responded on a scale where 1 = always difficult, 2 = usually difficult, 3 = sometimes difficult, 4 = usually easy, and 5 = always easy [17].

To calculate the mean score for each domain, the item scores were added together, multiplied by the total number of items within the domain, and then divided by the maximum possible score for that domain [19]. Participants who scored at or above the mean score for each domain of the HLQ were categorised as having 'high Health Literacy Level (HLL)', while those who scored below the mean score were classified as having 'low HLL'.

There is no single total HL score, as the multidimensional nature of the construct is better represented by separate scores for each of the nine domains. These domains are conceptually distinct, and separate scores provide insights into the strengths and limitations of the respondents [17, 20]. The Nepali version of the HLQ, along with socio-demographic and health-related information sources (healthcare providers, internet, books, family, friends, television, radio), was used to collect the information from the participants. The Cronbach's alpha for each domain was more than 0.8, indicating a higher level of internal consistency among the items in each domain. Data from the respondents were collected through a paper-based structured questionnaire via face-to-face interviews conducted by the trained enumerators.

## Statistical analysis

The collected data were initially entered in Excel and subsequently imported into IBM SPSS version 26 for further analysis. Both descriptive and inferential analyses were performed to gain insights from the dataset. Descriptive analysis was conducted to determine the frequencies and percentages of participants' sociodemographic characteristics. The normality of the data was assessed using the Shapiro-Wilk test. Multicollinearity tests were performed for all independent variables before data analysis. The Variance Inflation Factor (VIF) was <10, and the tolerance test yielded values less than one for each independent variable. The nine domains of the HLQ were treated as dependent variables. At the same time; age, gender, educational status, religion, ethnicity, occupation, economic status, residence, chronic disease profile, presence of co-morbidity, complications arising from NCDs, involvement in social activities, regular engagement in physical activities, attendance at health-related seminars, and social connectedness were considered as independent variables. The association between categorical HL score and categorical independent variables was assessed using the Chi-square test at the 95% significance level. Independent variables that demonstrated statistical significance were subsequently included in binary logistic regression. The multivariate logistic regression model fitted those variables with p-values less than 0.1 in the bivariate analysis.

## Ethical approval

Ethical approval was obtained from the Institutional Review Committee (Ref:329(6–11)E2) of the Institute of Medicine, Tribhuvan University, Nepal. Participants were informed that their participation was voluntary and assured of their anonymity and the confidential treatment of their responses before consent. A license to administer HLQ was obtained from Swinburne University of Technology.

# Results

A total of 303 participants attending the outpatient department consented to participate, resulting in a response rate of 90%.

## Socio-demographic characteristics

A total of 303 patients attending the outpatient department were included in the study. The mean ±SD age of the respondent was 47.4±16.18 years, with 56.1% female. A notable portion had higher education (26.1%), while 27.4% were illiterate. The majority of patients were Hindu (86.1%), with 29% belonging to the 'Chhetri' ethnic group. Most patients were homemakers (35.3%), and 72.3% were from the urban municipality, as presented in Table 1. (S1 File).

## Medical history

About the medical history of the patients, 35% had comorbidities, hypertension (31.4%) and diabetes (26.7%) being the most prevalent conditions. The majority (69.6%) engaged in social activities, and 65% participated in regular physical activities. Additionally, 86.5% reported having social connections and interpersonal relationships, as presented in Table 1.

## Sources of health information

Internet was the primary source of health information for most patients (48.7%), followed by family members (39.4%), with only 31.1% obtaining health information from healthcare providers, as depicted in Fig 1.

**Table 1. Socio-demographic and medical characteristics of the respondents (n = 303).**

| Characteristics | Category | Number (%) |
|---|---|---|
| Age | 18–44 | 121(39.9) |
| | 45–64 | 137 (45.2) |
| | 65 and above | 45 (14.9) |
| Gender | Female | 170 (56.1) |
| | Male | 133 (43.9) |
| Educational status | Illiterate | 83 (27.4) |
| | Basic level | 72 (23.8) |
| | Secondary level | 69 (22.8) |
| | Higher education | 79 (26.1) |
| Marital status | Married | 258 (85.2) |
| | Unmarried | 45 (14.8) |
| Religion | Hindu | 261 (86.1) |
| | Buddhism | 29 (9.6) |
| | Others | 13(4.3) |
| Ethnicity | Chhetri | 88 (29) |
| | Brahmin | 86 (28.4) |
| | Janajati | 85 (28.1) |
| | Others* | 44 (14.5) |
| Occupation | Housewife | 107 (35.3) |
| | Unemployed | 38 (12.5) |
| | Employed | 67 (22.1) |
| | Agriculture | 43 (14.2) |
| | Others | 48 (15.8) |
| Economic status | Below poverty line (< $ 2.15 USD/day) | 31 (10.2) |
| | Above poverty line (> $ 2.15 USD/day) | 272 (89.8) |
| Residence | Rural municipality | 84 (27.7) |
| | Municipality | 219 (72.3) |
| Type of NCDs | CVDs | 36 (11.9) |
| | Diabetes | 81 (26.7) |
| | Hypertension | 95 (31.4) |
| | Asthma | 62 (20.5) |
| | Others** | 78 (25.7) |
| Presence of co-morbidity | Yes | 106 (35) |
| Complications from NCDs | Yes | 84 (27.7) |
| Involving in social activities | Yes | 211 (69.6) |
| Regular exercise | Yes | 197 (65) |
| Attending health related seminars | Yes | 119 (39.3) |
| Social connectedness | Yes | 262 (86.5) |

Others* Tharu, Dalit, Thakuri; Others** Cancer, COPD, Epilepsy

## Health literacy level

The first five domains exhibited high HLL. The highest HLL was observed for 'Appraisal of health information' (79.5%), while the lowest was for 'Feeling understood and supported by healthcare providers' (60.4%). For the last four domains, the highest HLL was for 'Ability to

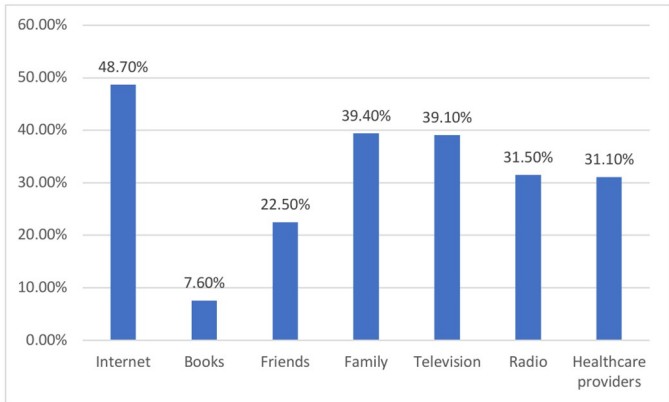

**Fig 1. Distribution of sources of health information.**

actively engage with healthcare providers' (60.7%), while 'Navigating the healthcare system' (18.2%) had the lowest, as presented in Table 2.

## Factors associated with health literacy domains

Gender was significantly associated with 'Feeling understood and supported by healthcare providers'. Educational status was significantly associated with six domains of HL except 'Actively managing own health', 'Appraisal of health information', and 'Navigating the healthcare system'. Respondents with a higher education level were more likely to have higher HL than illiterate respondents: 6.19 (95%CI: 2.30, 16.64) times more likely to have higher HL for 'Feeling understood and supported by healthcare providers', 5.51(95%CI:2.04,14.98) times higher HL for 'Having sufficient information to manage their health', 5.62 (95%CI:2.15,14.76) times higher HL for 'Social support for health', 16.43 (95%CI:5.47,48.99) times higher HL for 'Ability to actively engage with healthcare providers', 38.79(95%CI:11.2,134) times higher HL for ability to 'Find good health information' and 14.84 (95%CI:5.07,43.54) times higher HL for 'Understand health information well enough to know what to do' than the illiterate respondents respectively as presented in Tables 3 and 4.

Co-morbidity status and social connectedness were also associated with specific health literacy domains. Respondents with no co-morbidities were 2.41 (95%CI: 1.05, 5.52) times more

**Table 2. Health literacy level of respondents across different domains of HLQ (n = 303).**

| HL domains | Mean (SD±) | HLL | |
|---|---|---|---|
| | | High HLL | Low HLL |
| 1. Feeling understood and supported by healthcare providers | 2.70 (0.477) | 60.4% | 39.6% |
| 2. Having sufficient information to manage my health | 2.78 (0.43) | 67.3% | 32.7% |
| 3. Actively managing own health | 2.71 (0.48) | 68% | 32% |
| 4. Social support for health | 2.69 (0.48) | 65.3% | 34.7% |
| 5. Appraisal of Health Information | 2.80 (0.42) | 79.5% | 20.5% |
| 6. Ability to actively engage with healthcare providers | 3.25 (0.80) | 60.7% | 39.3% |
| 7. Navigating the healthcare system | 2.69 (0.46) | 18.2% | 81.8% |
| 8. Ability to find good health information | 3.21 (0.91) | 58.7% | 41.3% |
| 9. Understand health information well enough to know what to do | 3.13 (0.88) | 55.8% | 44.2% |

**Table 3. Multivariate logistic regression for factors associated with HL domain 1 to 5.**

| Variables | Category | Domain 1 | Domain 2 | Domain 3 | Domain 4 | Domain 5 |
|---|---|---|---|---|---|---|
| | | AOR# (95%CI) | AOR# (95%CI) | AOR# (95%CI) | AOR# (95%CI) | AOR# (95%CI) |
| Age group | 18–44 | 1 | 1 | 1 | 1 | 1 |
| | 45–64 | 0.74(0.37,1.49) | 1.42(0.68,2.93) | 0.85(0.42,1.73) | 1.33(0.66,2.64) | 1.87(0.80,4.38) |
| | ≥65 | 0.33(0.12,0.88)* | 0.82(0.31,2.19) | 0.49(0.19,1.28) | 1.46(0.57,3.77) | 1.16(0.38,3.50) |
| Gender | Male | 1 | 1 | 1 | 1 | 1 |
| | Female | 0.45(0.22,0.92)* | 0.99(0.48,2.03) | 0.74(0.36,1.28) | 1.24(0.62,2.50) | 0.68(0.29,1.60) |
| Educational status | Illiterate | 1 | 1 | 1 | 1 | 1 |
| | Basic level | 3.33(1.52,7.3)** | 3.85(1.72,8.62)** | 0.90(0.41,1.95) | 2.63(1.24,5.59)** | 2.15(0.82,5.63) |
| | Secondary level | 2.48(1.07,5.75)* | 4.53(1.86,11.00)** | 1.29(0.54,3.08) | 5.33(2.2,12.8)** | 1.27(0.47,3.44) |
| | Higher education | 6.18(2.30,16.64)** | 5.5(2.03,14.8)** | 2.01(0.73,5.52) | 5.6(2.15,14.6)** | 2.69(0.83,8.68) |
| Occupation | Housewife | 1 | 1 | 1 | 1 | 1 |
| | Employed | 0.94(0.37,2.39) | 0.73(0.28,1.89) | 1.66(0.62,4.39) | 1.49(0.58,3.82) | 1.33(0.43,4.12) |
| | Agriculture | 0.72(0.28,1.88) | 1.20(0.46,3.16) | 0.96(0.38,2.43) | 1.27(0.50,3.27) | 1.01(0.34,2.94) |
| | Others | 1.61(0.71,3.62) | 1.61(0.70,3.68) | 1.62(0.73,3.58) | 1.39(0.64,3.04) | 1.96(0.79,4.90) |
| Residence | Rural | 1 | 1 | 1 | 1 | 1 |
| | Urban | 1.78(0.97,3.31) | 1.51(0.81,2.81) | 1.42(0.78,2.61) | 1.08(0.59,1.97) | 0.99(0.49,2.00) |
| Co-morbidity | Yes | 1 | 1 | 1 | 1 | 1 |
| | No | 0.80(0.44,1.46) | 1.09(0.59,1.96) | 1.21(0.68,2.17) | 1.35(0.77,2.39) | 1.67(0.88,3.24) |
| Involving in social activities | Yes | 1 | 1 | 1 | 1 | 1 |
| | No | 0.81(0.42,1.55) | 2.24(1.09,4.59) | 1.57(0.87,3.06) | 1.08(0.57,2.06) | 1.52(0.70,3.30) |
| Regular physical activity | Yes | 1 | 1 | 1 | 1 | 1 |
| | No | 0.45(0.24,0.82)* | 0.51(0.28,0.92)* | 0.56(0.31,1.01) | 0.57(0.34,1.23) | 0.36(0.19,0.70) |
| Attending health related seminar | Yes | 1 | 1 | 1 | 1 | 1 |
| | No | 0.82(0.43,1.55) | 0.36(0.18,0.71)** | 0.43(0.22,0.84)* | 0.65(0.34,1.23) | 0.51(0.23,1.15) |
| Social connectedness | Yes | 1 | 1 | 1 | 1 | 1 |
| | No | 0.45(0.18,1.11) | 0.45(0.18,1.09) | 0.35(0.15,0.82)* | 0.61(0.26,1.42) | 0.37(0.15,0.92)* |

Domain1, Feeling understood and supported by healthcare providers; Domain2, Having sufficient information to manage my health; Domain3, Actively managing own health; Domain4, Social support for health; Domain5, Appraisal of Health Information

#AOR: Adjusted Odds Ratio,

*significantly associated at <0.05,

**significantly associated at <0.01,

1-Reference category

likely to have higher HL for 'Navigating the healthcare system'. Compared with the respondents having social connectedness, those without social connection had 0.35 (95% CI: 0.15, 0.82) times lower HL for 'Actively managing own health' and 0.38 (95% CI: 0.15, 0.93) times lower HL for 'Appraisal of health information'.

Compared to respondents who had attended a health-related seminar, respondents who had not participated had 0.36 (95%CI: 0.18, 0.72) times lower HL for 'Having sufficient information to manage their health', 0.43 (95%CI: 0.22, 0.84) times lower HL for 'Social support for health', and 0.36 (95%CI: 0.17, 0.73) times lower HL for 'Ability to find good information', as represented in Tables 3 and 4.

## Discussion

This study assessed the HLL and uncovered key factors linked to varying levels of HL among patients with NCD visiting the tertiary level of health care in Nepal. Our study identified that

**Table 4. Multivariate logistic regression for factors associated with HL domain 6 to 9.**

| Variables | Category | Domain 6 AOR# (95%CI) | Domain 7 AOR# (95%CI) | Domain 8 AOR# (95%CI) | Domain 9 AOR# (95%CI) |
|---|---|---|---|---|---|
| Age group | 18–44 | 1 | 1 | 1 | 1 |
| | 45–64 | 1.05(0.53,2.11) | 1.02(0.45,2.31) | 0.55(0.26,1.19) | 0.63(0.31,1.29) |
| | ≥65 | 1.29(0.49,3.39) | 2.12(0.63,7.06) | 0.63(0.21,1.83) | 0.33(0.11,0.94) |
| Gender | Male | 1 | 1 | 1 | 1 |
| | Female | 0.61(0.29,1.26) | 0.81(0.36,1.82) | 0.51(0.23,1.15) | 0.55(0.26,1.15) |
| Educational status | Illiterate | 1 | 1 | 1 | 1 |
| | Basic level | 2.82(1.31,6.05)** | 0.87(0.26,2.85) | 5.47(2.25,13.3)** | 2.65(1.16,6.04)** |
| | Secondary level | 4.20(1.78,9.89)** | 1.77(0.54,5.74) | 17(6.22,46.6)** | 6.79(2.75,16.7)** |
| | Higher education | 16.4(5.47,48.9)** | 3.18(0.95,10.6) | 38.79(11.2,134)** | 14.8(5.06,43.5)** |
| Occupation | Housewife | 1 | 1 | 1 | 1 |
| | Employed | 1.03(0.39,2.74) | 2.15(0.76,6.06) | 1.33(0.44,4.00) | 1.15(0.43,3.12) |
| | Agriculture | 0.48(0.18,1.24) | 0.37(0.07,2.06) | 0.86(0.31,2.41) | 0.73(0.27,1.96) |
| | Others | 1.08(0.48,2.42) | 0.99(0.36,2.71) | 1.01(0.40,2.50) | 0.93(0.39,2.20) |
| Residence | Rural | 1 | 1 | 1 | 1 |
| | Urban | 1.16(0.63,2.14) | 0.86(0.39,2.71) | 1.59(0.79,3.17) | 1.08(0.56,2.06) |
| Co-morbidity | Yes | 1 | 1 | 1 | 1 |
| | No | 1.59(0.89,2.84) | 2.41(1.04,5.55* | 0.96(0.49,1.85) | 1.52(0.82,2.83) |
| Involving in social activities | Yes | 1 | 1 | 1 | 1 |
| | No | 0.80(0.42,1.53) | 1.04(0.47,2.31) | 0.63(0.30,1.31) | 0.93(0.46,1.88) |
| Regular physical activity | Yes | 1 | 1 | 1 | 1 |
| | No | 0.70(0.38,1.27) | 0.65(0.29,1.43) | 0.75(0.38,1.49) | 0.79(0.41,1.51) |
| Attending health related seminar | Yes | 1 | 1 | 1 | 1 |
| | No | 0.61(0.32,1.17) | 0.89(0.42,1.88) | 0.35(0.17,0.72)** | 0.57(0.29,1.11) |
| Social connectedness | Yes | 1 | 1 | 1 | 1 |
| | No | 0.80(0.33,1.94) | 0.16(0.02,1.34) | 2.04(0.75,5.54) | 0.40(0.14,1.12) |

Domain6, Ability to actively engage with healthcare providers; Domain7, Navigating the healthcare system; Domain8, Ability to find good health information;

Domain9, Understand health information well enough to know what to do.

# AOR: Adjusted Odds Ratio,

* significantly associated at <0.05,

** significantly associated at <0.01,

1-Reference category

the patients with NCDs had more than half of high HLL for all the domains of HLQ except the domain 'Navigating the healthcare system'. The major findings revealed that educational attainment, regular physical activity, participation in health-related seminars, and a strong sense of social connectedness emerged as positive contributors to higher HLL across various domains. Conversely, comorbidities exhibited a negative association with HL, underscoring the intricate interplay between personal factors and HL outcomes.

Our study identified that the patients with NCDs scored high HLL for all the domains of HLQ except the domain 'Navigating the healthcare system'. These findings reflected that they excel in understanding health information, establishing trust with healthcare providers, ensuring understanding and making informed decisions. They are skilled at identifying reliable information and can resolve any conflicting information [17]. The scores for each domain provide a clear picture of where individuals with NCDs may need help, making it easier to identify

areas where support can be offered. These insights contribute to the comprehension and application of health information, thereby promoting practices that lead to good health [19]. The lower HL score for 'Navigating the healthcare system' might suggest that the individuals face challenges in advocating for themselves and struggle to navigate the healthcare system to address their health needs [17].

There is constant evidence of a relationship between educational status and HL [15, 21–24]. This study identified that the educational status of the study participant is significantly associated with all the domains of HLQ except domains 'Actively managing own health', 'Appraisal of health information' and 'Navigating the healthcare system'. Our study reported higher odds of HLL among individuals with higher educational levels for domains 'Actively engaging with healthcare provider' and 'Understanding health information', consistent with the findings from the studies in Ethiopia [19] and Denmark [25] conducted among patients with chronic disease. This implies that the respondents with higher educational attainment had higher abilities to assess and evaluate health systems and information. They have a higher level of trust, better communication and shared treatment decisions with health service providers [26]. They tend to be more informed about health-related hazards and prevention programs, improving their overall HL [27]. In contrast, respondents with lower educational attainment were likely to engage in unhealthy behaviours [24] and had delayed access to healthcare services, leading to poorer health outcomes [28].

The respondents above the age of 65 years were 67% less likely to have high HLL as compared with the respondents below 45 years for the domain 'Feeling understood and supported by healthcare providers', which is consistent with the findings of the study in Europe [15], and in remote Australia which revealed HLL was strongly associated with all the domains of HLQ for respondents less than 55 years [8]. The possible explanation could be age-related cognitive changes brought on by ageing and the increasing prevalence of sensory abnormalities, making it harder for older persons to communicate effectively with healthcare personnel [29]. The other possible explanation might be the dependence of older patients on their younger family members to seek health advice and services from healthcare providers. This could limit the direct communication between older patients with NCDs and health care providers.

The majority of the respondents in our study had limited HLL for 'Navigating the healthcare system', which is consistent with the study conducted in Australia among individuals with chronic health conditions [30]. This reflects the complex healthcare system, poorer user-friendly information, and poorer skills of the patients to interact and negotiate with health organisations and health professionals in shared decision-making [31, 32]. Mobilising the patient navigators to support patients in finding their way through the health and social care system could be the strategy to address this problem [33]. Our study revealed that respondents with no comorbidity had higher odds of HLL than those with comorbidity for 'Navigating the healthcare system'. This indicates people facing multiple health issues struggle with healthcare access due to communication problems, complex needs, and system issues [32]. Comorbid patients, often more dependent than non-comorbid individuals, face logistical and financial burdens that impede the caregiver and patient's ability to interact appropriately with the healthcare system [34]. These obstacles could make it harder for them to comprehend and utilise health-related information, leading to a lower HL [35]. These further necessitate a more accessible system that supports patients in making health decisions.

The respondents who were not involved in regular physical activities were 55% less likely to have a higher HLL than those engaged in regular physical activities for the domain 'Feeling understood and supported by healthcare provider'. This is in line with the findings from a similar study, which indicated that the respondents with regular physical activity had higher HLL [36]. The respondents involved in regular physical activities understand disease prevention

and the health system [37]. They adopt healthy behaviours that could improve cognitive functioning and decrease the rate of cognitive decline [29], enabling them to understand and explore health-related information.

In our study, the respondents who had not attended health-related seminars were less likely to have higher HLL than those who had participated in health seminars. Those respondents who had not attended health seminars were 64% less likely to have a higher HLL for the domain 'Having sufficient information to manage my health', 57% less likely to have a higher HLL for the domain 'Actively managing own health' and 65% less likely to have higher HLL for domain 'Ability to find good health information', respectively. This implies that the individuals participating in health seminars and conferences share new scientific information with doctors, talk about their experiences, and become more involved in their care [38]. This helps to improve their HL and leads to better health results. Such activities should be promoted to uplift the HL of patients with NCDs.

The individuals without social connection were 65% less likely to have a high HLL for the domain 'Actively managing own health' and 63% less likely to have a high HLL for the domain 'Appraisal of health information'. This indicates that limited social connections correlate with reduced information sharing and community engagement, negatively impacting HL, which is concordant with studies conducted in the USA and China [39, 40]. The respondents with high social life and connections frequently interacted with other people and health organisations; they exchanged their thoughts and experiences, which could expand the HLL [41]. A cross-sectional study conducted by Joy Agner et al. indicates that individuals who name more staff for health assistance tend to rely less on external help for understanding medical information [42]. It shows the mutual benefit of the doctor-patient relationship. The results also offer important knowledge on HLL that healthcare professionals and patients can utilise to improve their ability to understand and work together in healthcare management.

## Strength and limitation

To our knowledge, this study is the first among patients with NCDs in Nepal using HLQ. Additionally, this study was conducted among follow-up patients with NCDs visiting a health facility, making it difficult to generalise for general people. However, it was broadly representative nationwide as it was conducted in a pioneering government hospital and higher referral centre. As the purposive sampling technique and face-to-face interview through structured questionnaire were applied, respondent and selection bias were possible. We tried to reduce this by using standard and validated tools and techniques.

## Implication of study

Several key implications emerged based on our study on HL among Nepalese patients with NCDs. Given the significant influence of educational attainment, targeted interventions should enhance educational experiences to bolster HL. Addressing the notably lower scores in 'Navigating the healthcare system' suggests the introduction of patient navigators or similar roles to guide patients effectively. Additionally, age-specific interventions are crucial, given the challenges older individuals face in healthcare interactions. Integrated care models for patients with multiple comorbidities can improve their HL and outcomes, which can be tested in future studies. Promoting physical activity and advancing social connections through community engagement initiatives are vital to enhancing HLLs. Overall, collaborative efforts among healthcare providers, policymakers, and educators are essential to develop evidence-based strategies, ensuring improved HL and outcomes for patients with NCDs in Nepal.

## Conclusions

This study reveals that patients with NCDs exhibit varying HLL across different domains of the HLQ. Factors such as higher education, physical activity, attendance at health-related seminars, and social connectedness correlate positively with HLL, while comorbidity is negatively associated. Despite limitations due to its cross-sectional and exploratory nature, the findings emphasise the need for exploring a comprehensive approach to addressing health literacy, particularly in high-risk populations. Transforming health institutions into health literacy-friendly environments by implementing the feedback system of patients towards the healthcare provider and health system is crucial, and this approach needs to be tested in future studies.

## Supporting information

**S1 File.**
(ZIP)

## Acknowledgments

We thank Global Health & Equity, Swinburne University of Technology, for providing access to the HLQ. We would like to acknowledge all the enumerators, patients and staff of TUTH for their cooperation during the data collection process.

## Author Contributions

**Conceptualization:** Hari Joshi, Buna Bhandari.

**Data curation:** Hari Joshi, Bhoj Raj Kalauni, Kiran Bhusal.

**Formal analysis:** Hari Joshi, Rabindra Bhandari.

**Methodology:** Hari Joshi, Bhoj Raj Kalauni, Kiran Bhusal, Aastha Subedi, Buna Bhandari.

**Resources:** Hari Joshi.

**Software:** Hari Joshi, Bhoj Raj Kalauni.

**Supervision:** Buna Bhandari.

**Validation:** Buna Bhandari.

**Visualization:** Hari Joshi, Buna Bhandari.

**Writing – original draft:** Hari Joshi, Bhoj Raj Kalauni, Kiran Bhusal, Aastha Subedi, Buna Bhandari.

**Writing – review & editing:** Hari Joshi, Bhoj Raj Kalauni, Kiran Bhusal, Rabindra Bhandari, Aastha Subedi, Buna Bhandari.

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
