## [Decision Letter · Decision Letter 0]

20 Mar 2024

PONE-D-24-05594Health literacy among patients with non-communicable diseases at a tertiary level hospital in Nepal- A cross sectional studyPLOS ONE

Dear Dr. Bhandari,

Thank you for submitting your manuscript to PLOS ONE. After careful consideration, we feel that it has merit but does not fully meet PLOS ONE’s publication criteria as it currently stands. Therefore, we invite you to submit a revised version of the manuscript that addresses the points raised during the review process.

We look forward to receiving your revised manuscript.

Kind regards,

Nimesh Lageju

Academic Editor

PLOS ONE

Journal Requirements:

2. In this instance it seems there may be acceptable restrictions in place that prevent the public sharing of your minimal data. However, in line with our goal of ensuring long-term data availability to all interested researchers, PLOS’ Data Policy states that authors cannot be the sole named individuals responsible for ensuring data access (http://journals.plos.org/plosone/s/data-availability#loc-acceptable-data-sharing-methods).

Reviewers' comments:

Reviewer's Responses to Questions

**Comments to the Author**

1. Is the manuscript technically sound, and do the data support the conclusions?

Reviewer #1: Yes

Reviewer #2: Yes

2. Has the statistical analysis been performed appropriately and rigorously? 

Reviewer #1: Yes

Reviewer #2: I Don't Know

3. Have the authors made all data underlying the findings in their manuscript fully available?

Reviewer #1: Yes

Reviewer #2: No

4. Is the manuscript presented in an intelligible fashion and written in standard English?

Reviewer #1: Yes

Reviewer #2: No

5. Review Comments to the Author

Reviewer #1: This study by Buna Bhandari, et al., assessed HL levels and related factors among 303 NCD patients who came for follow-up from December 2022 to February 2023 at Tribhuvan University Teaching Hospital (TUTH) in Nepal.

Data was collected via face-to-face interviews by the trained enumerators using a structured Health Literacy Questionnaire (HLQ) containing 44 items (containing nine domains).

Multivariate logistic regression analysis was performed using SPSS version 26, with statistical significance at 0.05, to determine the associated factors with HL. The mean ±SD age of the respondent was 47.4 ± 16.18 years. More than half of the respondents were female (56.1%) and had higher HL for all the domains of HL except ‘Navigating the healthcare system’.

Results revealed that educational status was significantly associated with six of nine HL domains. Co morbidity, attending the health-related seminar, regular physical activity, and social connectedness were associated with at least one of the domains of HL. This study identified the important factors of HL, such as socio-demographic and medical factors among NCD patients. The patients with NCDs had more than half of high HLL for all the domains of HLQ except the domain navigating the healthcare system. The major findings revealed that educational attainment, regular physical activity, participation in health-related seminars, and a strong sense of social connectedness emerged as positive contributors to higher HLL across various domains. Conversely, the presence of comorbidities exhibited a negative association with HL, underscoring the intricate interplay between personal factors and HL outcomes.

Dependent (outcome) variables: All the 9 domains of the HL Questionnaire

Independent (predictors) variables: age, gender, educational status, religion, ethnicity, occupation, economic status, residence, chronic disease profile, presence of co-morbidity, complications arise from NCDs, involvement in social activities, regular engagement in physical activities, attendance at health related seminars, and social connectedness

The authors made all data underlying the findings fully available.

The data was also analyzed using both descriptive and inferential statistics which were rigorous and appropriate.

Discussions of the results were robust, citing similar studies conducted both within and outside Kenya and the continent.

Conclusions are in line with the findings

Writing quality and clarity: Satisfactory

Other observations:

1. Limitations of the study: The authors did well to mention the limitations of the study, including recommendations for future research in this area

2. Inclusion/exclusion criteria clearly explained.

Reviewer #2: Comments on Manuscript

PONE D-24-05594

“ Health literacy among patients with non-communicable diseases at a tertiary level hospital in Nepal - A cross sectional study

General Comments:

This is a good attempt. However, the entire manuscript needs to be reviewed and revised. The use of english needs to be improved throughout the manuscript.

Abstract:

Will need to be reviewed after the manuscript is revised

Introduction:

Lines 67-76 - Paragraph needs to be revised.

English language needs to be improved, to clarify the information presented and make it more understandable to the reader.

Some words are used repeatedly (for example similar/similarly), however, it is unclear what is being compared.

Study Population:

English language needs improvement - sentence construction and consistency.

Suggest use “Patients with NCDs” rather than NCD patients.

Statistical Analysis:

Use of Acronymn (VIF) with no explanation provided. Language needs improvement - review paragraphing; improve readability and understanding

Results:

Lacks clarity. Needs thorough review. Language needs to be improved significantly. The text lacks clarity, and is difficult to read and understand.

Lines 153 156 - For example “most” and major are being used for percentages which are around a quarter (1/4) or just over a third (1/3) of the study population. In some cases the percentages are quite similar yet one is reported as being major.

Tables 1, 2 and 3 need to be reviewed and revised to improve the presentation, accuracy of the data and readability. The tables should be simple to read and interpret. The abbreviation AOR needs to be explained at the end of each Table.

The information in the text must reflect what is in each table. This is sometimes not the case. At times one decimal place is reported in presenting data, while at other times two decimal places are reported. There is little or no reference to the specific table when the results are reported in the text. The reader has to search for the data in the tables, which makes understanding difficult. Sometimes what is reported as results in the text cannot be found in the table/s.

Discussion

The discussion is difficult to follow and understand. The use of the english language is at times incomprehensible. Example: Lines 236-242

The language needs significant revision to improve readability and understanding.

Statements are made throughout the section without adequate provision of references. Example lines 236-238;245-247;254=256;265-267;267-269

Some sentences are lengthy and at other times difficult to understand. Example lines 297-300; 310-311

As far as is possible, use language throughout so the manuscript that is easy to understand

Abbreviations

Not all the abbreviations used in the manuscript are noted

6. PLOS authors have the option to publish the peer review history of their article (what does this mean?). If published, this will include your full peer review and any attached files.

Reviewer #1: **Yes: **Haruna Ismaila ADAMU, MBBS; MPH; PhD

Reviewer #2: **Yes: **Glennis Andall-Brereton

---

## [Author Response · Author response to Decision Letter 0]

15 May 2024

Dear Editor,

We thank the editor and reviewers for critically reviewing our manuscript, “Health Literacy among the Patients with Non-communicable Disease at a Tertiary Hospital in Nepal—A Cross-sectional Study,” and for your valuable comments. 

We appreciate the time and effort you and the reviewers spent in reviewing our work. We have carefully considered each comment and are grateful for the constructive feedback. Please find the point-by-point response to the reviewers’ comments below.

Response 

Reviewer #1: This study by Buna Bhandari et al. assessed HL levels and related factors among 303 NCD patients who came for follow-up from December 2022 to February 2023 at Tribhuvan University Teaching Hospital (TUTH) in Nepal. Data was collected via face-to-face interviews by the trained enumerators using a structured Health Literacy Questionnaire (HLQ) containing 44 items (containing nine domains). The authors made all data underlying the findings fully available. The results were discussed robustly, citing similar studies conducted both within and across the continent. The conclusions are in line with the findings. Writing quality and clarity: Satisfactory Other observations: Limitations of the study: The authors did well to mention the study’s limitations, including recommendations for future research in this area. Inclusion/exclusion criteria were clearly explained.

Response: We thank the reviewer for thoroughly reviewing our manuscript and appreciating our work. 

Reviewer 2 comments

General Comments:

This is a good attempt. However, the entire manuscript needs to be reviewed and revised, and the use of English needs to be improved throughout.

Response: We would like to thank the reviewer for the constructive comments. We have reviewed and revised the language and English throughout the manuscript. 

Comment 1.1 Abstract:

Will need to be reviewed after the manuscript is revised

Response: We have revised the abstract after revising the manuscript 

Introduction

Comment: Lines 67-76 - Paragraph needs to be revised. English language needs to be improved, to clarify the information presented and make it more understandable to the reader.

Some words are used repeatedly (for example similar/similarly), however, it is unclear what is being compared.

Response: Thank you for the feedback. The language has been refined to enhance clarity and readability. Complex sentences have been simplified, and the information is presented clearly. The repetition of words, such as ‘similar’ and ‘similarly’, has been minimised. The paragraph provides clearer transitions between studies and findings, helping the reader understand more effectively. (Line 24-56)

Study Population

Comment: English language needs improvement - sentence construction and consistency.

Suggest the use of “Patients with NCDs” rather than NCD patients.

Response: Thank you. The necessary improvements are made to ensure clarity, especially focusing on sentence construction and consistency. The use of ‘patient with NCDs’ instead of ‘NCD patients’ has been implemented throughout the revised manuscript. 

Statistical Analysis

Comment: Use of Acronym (VIF) with no explanation provided. Language needs improvement - review paragraphing; improve readability and understanding.

Response: The acronym VIF is explained in the manuscript. We improved the readability of the paragraph, making it understandable. (see Lines 114-116)

Results

Comment: Lacks clarity. Needs thorough review. Language needs to be improved significantly. The text lacks clarity, and is difficult to read and understand.

Lines 153 156 - For example “most” and major are being used for percentages which are around a quarter (1/4) or just over a third (1/3) of the study population. In some cases the percentages are quite similar yet one is reported as being major.

Response: The results section's paragraphs are revised to improve the language and clarity. The sentences are made more concise, and the information is presented clearer and more organised. The usage of terms such as ‘most’ and ‘major’ for the relatively small percentages (around a quarter or just over a third) of the study population have been removed, and the percentages are reported without any subjective qualifiers, leading to a more accurate representation of the data. In the revised paragraph, consistency in reporting percentages is maintained. (See Line 136-180)

Comment: Tables 1, 2 and 3 need to be reviewed and revised to improve the presentation, accuracy of the data and readability. The tables should be simple to read and interpret. The abbreviation AOR needs to be explained at the end of each Table.

Response: Thank you for your valuable feedback. We have reviewed and simplified the format to make it easier to understand. Additionally, explanations for the abbreviation ‘AOR’ are included at the end of each table.

Comment: The information in the text must reflect what is in each table. This is sometimes not the case. At times one decimal place is reported in presenting data, while at other times two decimal places are reported. There is little or no reference to the specific table when the results are reported in the text. The reader has to search for the data in the tables, which makes understanding difficult. Sometimes what is reported as results in the text cannot be found in the table/s.

Response: Thank you for bringing this point to our attention. The text is carefully reviewed to accurately reflect the data presented in each table and referenced accordingly in the text (Line 141, 146, 150, 156, 169, 180). The 2-place decimal digit is standardized throughout the manuscript for consistency.

Discussion

Comment: The discussion is difficult to follow and understand. The use of the English language is, at times, incomprehensible. Example: Lines 236-242

Response: Thank you. Lines 236-242 have been revised and improved in readability. The English language has also been revised and restructured, making the sentences clearer and more understandable throughout the manuscript.

Comment: The language needs significant revision to improve readability and understanding. Statements are made throughout the section without adequate provision of references. Example lines 236-238;245-247;254=256;265-267;267-269

Response: Thank you for highlighting the referencing part. The language used in the above-mentioned lines has been revised and restructured for better flow. The statements are expressed clearly and concisely with proper references. 

Comment: Some sentences are lengthy and, at other times, difficult to understand. Example lines 297-300; 310-311

Response: We have revised the lengthy sentences and restructured them to enhance their readability and understanding.

Comment: As far as is possible, use language throughout so the manuscript that is easy to understand.

Response: The language in the manuscript has been revised. We made necessary adjustments to enhance its readability while maintaining the integrity and accuracy of its content.

Abbreviations

Comment: Not all the abbreviations used in the manuscript are noted

Response: All the abbreviations used in the manuscripts are now listed in the manuscript.

We hope we can satisfactorily address the comments, and now our revised manuscript is suitable for publication in your esteemed journal. 

Thank you

---

## [Editor Report · Decision Letter 1]

20 May 2024

Health literacy among patients with non-communicable diseases at a tertiary level hospital in Nepal- A cross sectional study

PONE-D-24-05594R1

Dear Dr. Bhandari,

We’re pleased to inform you that your manuscript has been judged scientifically suitable for publication and will be formally accepted for publication once it meets all outstanding technical requirements.

Kind regards,

Nimesh Lageju

Academic Editor

PLOS ONE

---

## [Editor Report · Acceptance letter]

24 May 2024

PONE-D-24-05594R1 

PLOS ONE

Dear Dr. Bhandari, 

I'm pleased to inform you that your manuscript has been deemed suitable for publication in PLOS ONE. Congratulations! Your manuscript is now being handed over to our production team.

Kind regards, 

on behalf of

Dr. Nimesh Lageju 

Academic Editor

PLOS ONE